# Transcriptome and Metabolome Analysis of Selenium Treated Alfalfa Reveals Influence on Phenylpropanoid Biosynthesis to Enhance Growth

**DOI:** 10.3390/plants12102038

**Published:** 2023-05-19

**Authors:** Fengdan Wang, Jie Yang, Yi Hua, Kexin Wang, Yue Guo, Yalin Lu, Siqi Zhu, Pan Zhang, Guofu Hu

**Affiliations:** Department of Grassland Science, College of Animal Science and Technology, Northeast Agricultural University, Harbin 150030, China

**Keywords:** *Medicago sativa* L., sodium selenite, transcriptome, metabolome, phenylpropanoid biosynthesis

## Abstract

Selenium (Se) plays an important role in the growth of plants. Alfalfa (*Medicago sativa* L.) is a perennial legume forage crop with high nutritional value and Se-rich functions. Many studies have shown that selenium can promote alfalfa growth, but few have explored the molecular biology mechanisms behind this effect. In this study, alfalfa was divided into two groups. One group was sprayed with sodium selenite (Na_2_SeO_3_) and the other group was sprayed with distilled water as a control. This study determined the growth, reproductive traits, physiological changes, transcriptome and metabolome of both groups of alfalfa. We found that foliar spraying of 100 mg/L Na_2_SeO_3_ could significantly increase the growth rate, dry weight, total Se content, amount of pollen per flower, pollen viability, pod spirals, and seed number per pod of alfalfa plants. The level of chlorophyll, soluble protein, proline, and glutathione also increased dramatically in Na_2_SeO_3_-sprayed alfalfa seedlings. After transcriptome and metabolome analysis, a total of 614 differentially expressed genes (DEGs) and 1500 differentially expressed metabolites (DEMs), including 26 secondary differentially metabolites were identified. The DEGs were mainly enriched in MAPK signaling pathway, phenylpropanoid biosynthesis, isoflavonoid biosynthesis, cutin, suberine, and wax biosynthesis, and glycerolipid metabolism. The DEMs were mainly enriched in flavone and flavonol biosynthesis, carbon metabolism, glyoxylate and dicarboxylate metabolism, nitrogen metabolism, and phenylpropanoid biosynthesis. Integrative analysis of transcriptome and metabolome showed that the foliar spraying of Na_2_SeO_3_ mainly affects phenylpropanoid biosynthesis to promote alfalfa growth.

## 1. Introduction

Selenium (Se) boosts plant development and strengthens their tolerance to environmental variables and infections [1,2]. Studies have shown that foliar spraying of Se can increase antioxidant enzyme activity, reduce membrane damage, and decrease active oxygen content in soybean leaves [3]. In another study on selenium, researchers have found that a modest concentration of Se can protect cucumber, rape, and rice from abiotic stressors such as water deficiency, chilling, and arsenic toxicity [4,5]. In experiments on biological stress, Se has been found to significantly protect Indian mustard from fungal infections and herbivorous attacks by caterpillars [6,7]. 

Alfalfa (*Medicago sativa* L.) is a perennial leguminous forage with high nutritional value and Se-rich functions [8,9,10]. The application of selenium in alfalfa has been extensively studied, and these studies generally indicate that a modest concentration of Se promotes alfalfa growth. One study has indicated that 10 and 15 μmol·L^−1^ Se (elemental selenium) enhanced soluble sugars and starch accumulation in the shoots and roots of young plants. Meanwhile, Se up-regulates carbohydrate metabolism via altered redox potential which may have some stimulatory effects on nodulation [11]. In other studies, researchers found that soaking alfalfa (*Medicago sativa* L. cv.’pianguan’) seeds in a 0.5 mmol·L^−1^ sodium selenite (Na_2_SeO_3_) solution for 12 h, resulted in seedlings with significantly higher antioxidant ability than the control group [12]. In another study of alfalfa seeds, researchers found that selenium application mainly promoted the germination of alfalfa seeds by increasing sugar conversion efficiency [13]. In studies on alfalfa yield and nutrient quality, researchers found that spraying with nano-selenium and sodium selenite can increase alfalfa yield and protein content [14]. 

At the same time, Se-enriched forage is a risk-free and efficient solution to unequal Se intake [15]. For instance, adding Se-enriched alfalfa to the diet of shell-laying chickens can significantly increase the amount of DHA (docosahexaenoic acid), linolenic acid, and linoleic acid found in the egg yolk [16]. Se-enriched forage can boost the antibody titer of adult beef cattle, the Se content of whole blood, and the growth rate in weaned calves [17]. Enriching plants with Se during cultivation is an efficient method of producing Se-rich forage crops because plants increase the accumulation of Se and convert Se into organic selenium, such as methylselenocysteine [18]. The bioavailability of organic selenium is higher than that of inorganic selenium (about 1.5–2.0 times) and it is organic selenium that exerts its biological activity in animals and humans [19,20]. Indirectly increasing the consumption of Se in animals and people through the food chain can benefit health over the long run [21].

Although adding Se-rich fertilizers to the soil and applying Se sprays to plants are common methods for increasing the selenium content of crops [22], Se compounds are quickly fixed and lost in the soil, which limits the amount of Se that can be used by plants [23]. Therefore, foliar spraying with Se is advantageous for increasing the utilization rate of Se and is an effective method for regulating the Se content of plants [24]. This is why we used foliar spraying in this study.

With the development of sequencing technology, high-throughput omics such as transcriptome and metabolome have been widely used. Integrated analysis of the transcriptome and metabolome is an effective method for exploring biological problems at both the “cause” and “effect” levels simultaneously, and for verifying each other. Differentially expressed genes and metabolites have been combined and analyzed according to distinct physiological, pathological, growth, and developmental characteristics to discover critical metabolic pathways and genes and explain the regulatory mechanism. For example, researchers’ integrative analysis of the metabolome and transcriptome revealed the phosphate deficiency response pathways of alfalfa. This study found that Pi deficiency was summarized as local systemic signaling pathways, including root growth, stress-related responses consisting of enzymatic and nonenzymatic systems, and hormone signaling and systemic signaling pathways including Pi recycling and Pi sensing in the whole plant, as well as Pi recovery, and nitrate and metal absorption in the roots [25]. In another study, researchers compared the physiological and transcriptomic responses of two alfalfa varieties with different salt tolerance levels to reveal their salt tolerance mechanisms. They found that the highly salt-tolerant alfalfa employed various strategies to cope with salinity conditions by regulating ionic homeostasis, antioxidative enzyme activities, and fatty acid metabolism at both the transcriptional and physiological levels [26].

Although numerous studies have demonstrated that selenium promotes alfalfa growth, the underlying molecular mechanisms remain largely unexplored. In this study, we aim to elucidate the relationship between alfalfa and selenium by examining the growth, reproductive traits, physiological changes, transcriptome, and metabolome profiles of alfalfa treated with sodium selenite (Na_2_SeO_3_). Our findings will provide a theoretical foundation for further investigation into selenium enrichment in alfalfa and other plants.

## 2. Results

### 2.1. Effects of Na_2_SeO_3_ Spraying on Alfalfa Growth and Reproduction

The growth rate increased dramatically on the first day after being sprayed with Na_2_SeO_3_ (ST), which was 1.67 times that of CK (control group). However, no growth was promoted in the subsequent seven days (Figure 1a). Spraying Na_2_SeO_3_ on alfalfa at the early flowering stage did not influence plant height and FW (fresh weight) but had a significant effect on DW (dry weight) and FW/DW ratio compared to CK (*p*-value < 0.05, Table 1). The FW/DW ratio of ST declined by 1.72%, but the DW increased by 61.11%. ST had a considerably higher total Se concentration than CK (*p*-value < 0.05). Spraying Na_2_SeO_3_ did not affect the number of florets on single reproductive branches and the weight of 1000 grains of alfalfa (Table 2), but it significantly enhanced the total amount of pollen (per flower) and pollen viability, the turns of pod spirals, and the number of seeds per pod (*p*-value < 0.05). In ST, the amount of pollen, pollen viability, pod spirals, and seed count per pod increased by 43.17%, 20%, 16.81%, and 34.48%, respectively, compared to CK.

### 2.2. Effects of Na_2_SeO_3_ Spraying on Physiological Changes of Alfalfa

The concentration of chlorophyll a and b in the ST increased substantially by 27.03% and 30.49% relative to CK at 6 h (*p*-value < 0.05) after spraying Na_2_SeO_3_ and peaked at 12 h, increasing by 28.48% and 33.03%, respectively (Figure 1b,c). At 48 h, the concentration of chlorophyll a and b showed a decreasing tendency and was somewhat lower than that of CK. The content of SP (soluble protein) in ST was substantially greater than in CK at 12, 24, and 48 h (*p*-value < 0.05) and peaked at 215.19 μg·g^−1^FW, which was 20.26% higher than CK at 48 h. (Figure 1d). At 6 and 48 h after Na_2_SeO_3_ spraying, the Pro (free proline) content in ST was substantially higher than that in CK (Figure 1e). The content of GSH (reduced glutathione) in ST substantially rose after 6 h, peaked at 12 h at 510.93 μg·g^−1^FW, and increased by 21.77% when compared to CK (Figure 1f).

### 2.3. Transcriptomic Analysis of Alfalfa Sprayed with Na_2_SeO_3_

The alfalfa tissues sampled at 0 and 6 h in ST and CK were used to construct cDNA libraries and sequence the transcripts with an Illumina HiSeq4000. The raw reads ranged from 43,909,580 to 52,649,430, while the amount of sequencing data was between 6.59 G and 7.90 G (Table 3). After filtration, the range of clean readings was between 43,074,702 and 51,661,074, the amount of data was between 6.46 G and 7.75 G, and the average GC content was 42.83%. The alignment rate ranged from 73.47 to 78.33 percent when HISAT (hierarchical indexing for spliced alignment of transcripts) was employed to perform reference genome alignment on preprocessed valid data to extract the location information and specific sequence information of clean reads. |log_2_FC| ≥ 1 and *p*-value < 0.05 were used to identify DEGs, and a total of 614 DEGs, comprising 278 up-regulated and 336 down-regulated genes, were identified (Appendix A).

GO (Gene Ontology) function enrichment resulted in the assignment of 810 functions to 551 DEGs. There were 69 DEGs significantly enriched in biological processes, 12 in cellular components, and 54 in molecular activities. “Protein phosphorylation” was the most enriched term in the biological process category. The most enriched term in molecular function was “protein serine/threonine kinase activity”. The most enriched term for the cellular component was “apoplast” (Figure 2a). Among the 614 DEGs, 88 KEGG (Kyoto Encyclopedia of Genes and Genomes) pathways were enriched for 230 genes. These DEGs were predominantly associated with “MAPK signaling pathway-plant”, “Phenylpropanoid biosynthesis”, “Cysteine and methionine metabolism”, “Glycerolipid metabolism”, “Glycolysis/Gluconeogenesis”, “Isoflavonoid biosynthesis”, “Cutin, suberine, and wax biosynthesis”, and “Stilbenoid, diarylheptanoid, and gingerol biosynthesis” (Figure 2b). To evaluate the quality and reproducibility of the sequencing results, eight DEGs were selected at random for qRT-PCR validation using alfalfa GAPDH as the internal reference gene (Appendix A). The qRT-PCR results for the differentially expressed genes agreed with the sequencing data (Appendix A).

### 2.4. Metabolomic Analysis of Alfalfa Sprayed with Na_2_SeO_3_

In this study, we used untargeted metabolomics to analyze ST and CK. The total ion flow chromatogram of all samples showed that the alignment effect of the original peak was good, and all samples had a strong signal, large peak capacity, and good reproducibility, indicating that the result was reliable (Appendix A). Principal component analysis (PCA) reveals that the quality control samples (QC) represented by the orange origin in the middle overlap and cluster, indicating that the test data are highly reliable (Appendix A). There was a clear distinction between ST and CK, which may indicate substantial differences in metabolic processes.

A total of 1500 significantly different expressed metabolites (DEMs) were identified in alfalfa tissues sprayed with Na_2_SeO_3_ after comparing ST and CK, including 1051 up-regulated DEMs and 449 down-regulated DEMs (Appendix A). In addition, 26 secondary DEMs were identified, of which 14 were up-regulated and 12 were down-regulated (Figure 3a). These metabolites were involved in 29 KEGG pathways, including “Carbon metabolism”, “Flavone and flavonol biosynthesis”, “Glyoxylate and dicarboxylate metabolism”, “Nitrogen metabolism”, and “phenylpropanoid biosynthesis” (Figure 3b).

### 2.5. Integrative Analysis of Transcriptomics and Metabolomics Involved in Na_2_SeO_3_ Promotion of Alfalfa Growth

Integrative analysis of DEGs and DEMs revealed that 101 DEGs were substantially related to 60 DEMs, with 41 up-regulated DEGs, 60 down-regulated DEGs, 27 up-regulated DEMs, and 33 down-regulated DEMs (Appendix A). Following correlation analysis, these DEGs and DEMs were enriched for 249 GO terms (Figure 4a). There were 120 DEGs engaged in biological processes, including the oxidation–reduction process, response to wounding, defense response to fungus, and response to oxidative stress, of which 60 DEGs had a *p*-value < 0.05. There were 33 DEGs implicated in cellular components, with 9 DEGs having a *p*-value < 0.05, comprising apoplast, cell periphery, plant-type vacuole, and extracellular region. There were 95 DEGs implicated in the molecular function, and 56 DEGs had a *p*-value < 0.05, including metal ion binding, peroxidase activity, ATPase activity, chitinase activity, and dioxygenase activity. The most enriched KEGG pathway, phenylpropanoid biosynthesis, involved 51 of these 105 DEGs (Figure 4b).

### 2.6. Characterization of the Phenylpropanoid Biosynthesis Pathway

To explore the potential function of the phenylpropanoid biosynthesis pathway, which had the largest number of DEGs in response to Na_2_SeO_3_ spraying, we characterized the DEGs and DEMs involved in the phenylpropanoid biosynthesis pathway (Appendix A). There were 17 DEGs in the phenylpropanoid biosynthesis pathway, of which 7 DEGs were up-regulated and 10 DEGs were down-regulated (Appendix A). There were 14 DEMs in the phenylpropanoid biosynthesis pathway, of which 7 DEMs were up-regulated and 7 DEMs were down-regulated (Appendix A). These DEMs can be divided into four categories: amino acid metabolism, biosynthesis of other secondary metabolites, membrane transport, and metabolism of cofactors and vitamins. The DEGs enriched in this pathway were mapped on eight chromosomes, and the highest proportion was located on chromosome 3 (four DEGs, Appendix A). The length of 17 DEGs varied and ranged from 97 amino acids to 536 amino acids. The theoretical molecular weight ranged from 10,228.32 u to 58,955.57 u, and the isoelectric point, instability index, and aliphatic indexes were between 5.01 and 9.07, 11.77 and 44.68, and 6.39 and 103.06, respectively. Except for one hydrophobic protein, the proteins encoded by the other 16 DEGs, all had negative values, indicating that they are all hydrophilic proteins. There were 11 DEGs that had untranslated regions, and 5 DEGs only had CDS (Figure 5a). Twenty different motifs were predicted, and the higher number of different motifs that a gene contained was nine (Figure 5b).

## 3. Discussion

### 3.1. Effect of Growth and Reproductive Traits of Alfalfa

Herein, we found that spraying Na_2_SeO_3_ caused significant improvements in DW, FW/DW ratio, and total Se content in ST compared to CK (Table 1). The optimal Se concentration can stimulate plant growth and substantially increase dry matter in vegetative organs [27,28]. For instance, the application of Na_2_SeO_3_ can enhance the photosynthetic capacity and development of tomato seedlings under salt stress [29]. In the case of adapting to the application of selenium concentration, chicory (*Cichorium intybus* L.) and ryegrass (*Lolium perenne* L.) also promoted growth and increased dry weight [30,31]. In this study, foliar spraying of Na_2_SeO_3_ increased plant height and growth rate considerably on the first day (Figure 1a). However, at the early flowering stage, the plant height did not increase significantly, which may be related to the accumulation of more dry matter in reproductive organs [32]. Se has a beneficial effect on the reproductive traits of plants. Exogenous Se might improve the flowering index of cucumber [33], ameliorate the decline in pollen and increase the number of pepper flowers [34]. It can also enhance pollen germination, pollen viability, and pollen tube growth in mung beans (*Vigna radiata*) [35]. In this experiment, relative to the control group, the experimental group demonstrated no significant variations in the number of florets and the weight of 1000-seed units. However, there were significant improvements observed in pollen viability, the total amount of pollen, the number of seeds per pod, and the number of turns in pod spirals (Table 2). The results indicate that Na_2_SeO_3_ promotes the reproductive traits of alfalfa to a certain extent.

### 3.2. Effect of Physiological Changes of Alfalfa

Chlorophyll is a vital substance in photosynthesis and the absorption component in the metabolic process of plants [36]. Se boosts the respiration rate of mitochondria and the electron transfer rate of chloroplasts, leading to an increase in chloroplast content [37]. By adding Se fertilizer to potted plants at an appropriate quantity, chlorophyll a, and b can be stimulated [38]. In this experiment, the concentrations of chlorophyll a and b initially increased and then declined (Figure 1b,c). Meanwhile, the metabolome data revealed that glutamate increased considerably at 6 h (Appendix A). The increase in chlorophyll content may result in a high concentration of glutamate in the plant, followed by a downward trend once the glutamate content decreases. SP is an essential nutrient and osmotic regulator within living organisms. The growth and accumulation of SP can enhance the ability of cells to retain water and defend their biofilm [39]. A suitable concentration of Se can increase the SP content of *Atractylodes macrocephala* Koidz seeds and seedlings [40]. According to the findings by Hu [41], spraying nano-selenium on the leaves of purple potatoes resulted in an initial increase and subsequent decrease in the soluble protein content. In the present experiment, a similar trend was observed in the soluble protein content of alfalfa seedlings in the ST group compared to the control group (Figure 1d). This result is in alignment with Hu’s previous findings. Proline can be utilized as a signaling molecule to regulate mitochondrial activity, alter cell proliferation or death, and activate the production of specific genes, which is crucial for plant growth. Another study found that Se sprayed on wheat leaves greatly boosted the free proline content [42]. In this experiment, the free proline concentration in ST increased dramatically at 6 h, then declined and increased with time (Figure 1e), following metabolome profiling, we observed an upregulation in the concentration of glutamate in the phenylpropanoid biosynthesis pathway at the 6-h mark post-treatment (Appendix A). Based on these results, we postulate that the elevated levels of proline observed may be attributable to the increased availability of its precursor, glutamate, though further research is required to confirm this hypothesis. GSH is an antioxidant that helps prevent reactive oxygen species, such as free radicals and peroxides, from damaging vital cellular components. Zhao [43] found that adding Na_2_SeO_3_ to wheat significantly boosted its GSH content. At the same time, the same phenomenon also appeared in this experiment. In an earlier study, selenium has been shown to be an important component of glutathione peroxidase (GSH-PX); GSH-PX, uses GSH as a coenzyme to protect cells from oxidative stress [44]. This may be the reason why Na_2_SeO_3_ can alleviate some adversity.

### 3.3. Effect of Differentially Expressed Genes in Alfalfa

Among the 614 DEGs, 230 DEGs were enriched in 88 KEGG pathways, 8 of which had *p*-value < 0.05. These pathways can be categorized as signal transduction, secondary metabolite production, lipid metabolism, carbohydrate metabolism, and amino acid metabolism. The plant MAPK signaling pathway plays a central role in abiotic stress. The DEGs that regulate the potential protein phosphatases *2C24* and *2C56* in the MAPK signaling pathway were up-regulated in this experiment. Protein phosphatase is essential for maintaining the balance between reactive oxygen species (ROS) and phosphorylation. Pei [45] discovered that a low concentration of Se can activate protein phosphatase and the MAPK signaling pathway, which is consistent with this study.

The biosynthesis of secondary metabolites was primarily influenced by Se treatment for phenylpropanoid biosynthesis, isoflavones, stilbenes, diarylheptane, and gingerol. The application of Se can promote the activity of phenylalanine metabolism in Tartary buckwheat, thus promoting the formation of phenylpropanoid biosynthesis [46]. In the biosynthesis of phenylpropanoid biosynthesis, 4 DEGs were up-regulated, and 10 DEGs were down-regulated. Transcriptome analysis revealed a substantial enrichment of DEGs involved in phenylpropanoid production in Se-treated *Puccinellia tenuiflora* [47]. The biosynthesis of flavonoids, isoflavones, and anthocyanins in Se-rich Tartary buckwheat was mainly facilitated by phenylpropanoids [48]. Stilbene compounds can be used as plant protectants to boost peanut seedlings’ resistance to fungi [49].

The lipid metabolic pathway includes the biosynthesis of cutin, corky, and waxy, as well as glycerol ester metabolism. The DEGs involved in cutin, corky, and waxy biosynthesis were all down-regulated, while 4 DEGs involved in glycerol ester metabolism were up-regulated, and there were down-regulated DEGs in glyceride metabolism. Cutin is an insoluble polymer structural component of the cuticular layer of all aboveground plant parts except the periderm. Cork is a component of the cell walls of the endoderm and ectoderm cells of the cork layer [50]. Waxes are a type of organic substance that cover the outermost layer of plants and govern the passage of gases, water, and solutes, protect plants from biotic and abiotic stresses, and regulate plant morphology [46]. Se has an effect on glyceride metabolism in yeast cells [51]. Therefore, the down-regulation of genes may be associated with Se absorption and transport. Carbohydrate metabolism involves numerous physiological and biochemical processes that play a crucial role in synthesizing carbohydrates within organisms. 7 DEGs involved in the glycolysis/glycolysis/gluconeogenesis pathway of carbohydrate metabolism were up-regulated in this experiment, indicating that the energy accumulation required during the growth of alfalfa sprayed with Na_2_SeO_3_ accumulated primarily via photophosphorylation to oxidative phosphorylation [11]. Amino acids are the fundamental building blocks of biologically functioning macromolecular proteins and the necessary components of the proteins required for animal and plant sustenance. Se can transfer oxygen atoms of hydroxyl groups in the R group of serine to produce selenocysteine and sulfur atoms in methionine to produce selenomethionine [52,53]. In the transcriptome study of kale treated with Na_2_SeO_3_, DEGs were mainly focused on the metabolism of cysteine and methionine [54]. 7 DEGs were up-regulated and 1 DEG was down-regulated in cysteine and methionine metabolism, which may be due to Se replacing the substrate for the synthesis of cysteine and methionine being converted to selenomethionine, which further promotes the metabolism of cysteine and methionine in plants.

### 3.4. Effect of Differentially Metabolites of Alfalfa

In this paper, a total of 1500 DEMs were involved in 29 KEGG pathways, and the arginine biosynthesis, flavone and flavonol biosynthesis, and biosynthesis of phenylpropanoid biosynthesis pathways displayed considerable variances (Figure 3). Arginine is necessary for cell proliferation and division and affects the plant’s antioxidant system. Exogenous arginine administration significantly reduced tomato membrane lipid peroxidation under water pressure [55]. The glutamate metabolites in the arginine metabolic pathway were up-regulated in alfalfa sprayed with Na_2_SeO_3_ in this study. The foliar application of Na_2_SeO_3_ has the potential to stimulate the production of arginine precursors and significantly improve the quality of alfalfa. Flavonoids and phenylpropanoid molecules have great importance in plant growth, development, and response to stress [56,57]. Carrots treated with SeO_2_ had significantly higher levels of total phenols and flavonoids, as well as greater antioxidant capacity [58]. The metabolic pathway of phenylpropanoid in apples was greatly enhanced with the application of sodium nitroprusside to enhance the fruits’ resistance to Penicillium [59]. In this study, DEMs, including phenylpropanoid biosynthesis, flavonoids, and isoflavones, were significantly up-regulated following Na_2_SeO_3_ spraying. Consequently, the foliar application of Na_2_SeO_3_ might increase the accumulation of arginine and flavonoids, promoting the growth and development of alfalfa. 

### 3.5. Integrative Analysis of Transcriptomics and Metabolomics

There are sophisticated regulatory processes within the organism [60]. The integrative study revealed that 17 DEGs and 14 DEMs are involved in the pathway of phenylpropanoid biosynthesis. The most common matching enzyme was oxidoreductase, which may provide energy for synthesizing, metabolism, and transporting phenylpropanoid biosynthesis. In alfalfa, phenylpropanoid biosynthesis pathways produce flavonoids and isoflavones compounds that interact with helpful microbes (symbiotic flavonoid inducers of rhizobia) and guard against pathogens (flavonoids or isoflavones to defend against pathogens) [61,62]. Hence, the phenylpropanoid biosynthesis pathway is essential for alfalfa growth and development after Na_2_SeO_3_ spraying.

## 4. Materials and Methods

### 4.1. Experimental Site

The experiment was conducted at the Grassland Science Laboratory (Heilongjiang Province Key Laboratory) of Northeast Agricultural University (NEAU), China. Seed germination and seedling cultivation were carried out in the dark room and seedling rearing room in the laboratory (see Section 4.2 for details). Determination of growth, reproductive traits, and physiological changes was carried out in the laboratory (see Section 4.3 and Section 4.4 for details). The determination of transcriptome and metabolome was carried out by Lc-Bio Technologies (Hangzhou) Co., Ltd., Hangzhou, China (see Section 4.5, Section 4.6, Section 4.7 and Section 4.8 for details).

### 4.2. Plant Cultivation and Treatments

The alfalfa variety used in this study was ‘Dongnong No.1’(*Medicago sativa* L. cv. ‘Dongnong No. 1’). More detail on the parentage, year of release, yield potential, maturity group, and salient characteristics of Dongnong No.1 were described in the “Report of Breeding of New Variety of *Medicago sativa* L.cv. ‘Dongnong No. 1’” [63]. The seeds of alfalfa were supplied by the grassland science department and harvested in 2019. The seeds of alfalfa were sterilized for 10 min with 10% sodium hypochlorite (NaClO) and rinsed 4–5 times with distilled water. The seeds were then germinated in the dark room containing moist filter paper at 25 °C. Five days after germination, the seedlings were transplanted into plastic pots (20 cm length × 20 cm width × 20 cm height, the volume of the pot was 8 dm^3^) filled with vermiculite and perlite (1:1). There were 10 seedlings in each pot, and a total of 40 pots were used in this study. The pots were placed in a growth room with 60% humidity and a 16-h photoperiod at 24 °C and the seedlings were irrigated every two days with 1/2 Hoagland nutritional solution. Four weeks after being transplanted, the seedlings were separated into two groups. The treated group (ST) was sprayed with 100 mg/L sodium selenite (Na_2_SeO_3_), and the concentration of Na_2_SeO_3_ used to promote alfalfa growth was determined based on a preliminary experiment. In the preliminary experiment, we studied the effect of different concentrations of Na_2_SeO_3_ on MDA (malonaldehyde) content in alfalfa. We chose 100 mg/L as the concentration for this experiment, as it resulted in the lowest MDA content, indicating less or no damage to the plants. The control group (CK) was sprayed with distilled water. There were 20 pots in each group, and the standard spraying amount was determined by visually observing the appearance of water droplets on the surface of the leaves. The leaves were taken at 0, 6, 12, 24, 48, and 96 h after spraying and stored at −80 °C freezer. These samples were used to determine physiological changes and transcriptome. These tests were carried out with three separate biological replicates, with each sampling time involving the whole leaves of 5 alfalfa plants chosen randomly from 20 pots. 15 alfalfa plants were sampled each time, and the sampling has conducted a total of 6 times. The metabolome was carried out with 6 replicates. Surviving seedlings were allowed to continue developing to evaluate their development and reproductive characteristics. The period in days that the alfalfa survived was 57 days (from 24 October 2020 to 20 December 2020).

### 4.3. Determination of Growth and Reproductive Traits

After spraying with Na_2_SeO_3_, the growth rate was determined by measuring the absolute height of 10 alfalfa seedlings daily for one week. The plant height was obtained by measuring the absolute height of ten alfalfa seedlings at the early flowering stage. During the early flowering stage, the aboveground tissue biomass was also assessed. The fresh weight (FW) was obtained by measuring the aboveground tissues of harvested alfalfa, whereas the dry weight (DW) was established by drying the same tissues to a consistent weight in an oven. The tissues were then ground to determine the total Se content using inductively coupled plasma mass spectrometry (ICPMS) [64]. These studies were conducted with three separate replicates, and each replicate contained 10 randomly selected uniform plants. Given the significant promoting effect of selenium on the reproductive traits of plants [33,34,35], this study also measured the reproductive characteristics of alfalfa after spraying sodium selenite. The reproductive traits, including the number of florets and amount of pollen per flower, pollen viability, pod spirals, seed numbers per pod, and 1000-seed weight, were measured at the full flowering and podding stages. The vitality of the pollen was assessed using the in vitro germination method [65]. The amount of pollen per flower is observed using the dilution method. The rest of the reproductive traits were quantified using observational methods.

### 4.4. Determination of Physiological Changes

The aboveground tissues sampled at 0, 6, 12, 24, 48, and 96 h were used to determine the physiological changes of alfalfa after being sprayed with 100 mg/L Na_2_SeO_3_. The soluble protein (SP) content was determined using the Coomassie brilliant blue method [66], the reduced glutathione (GSH) content was determined using spectrophotometry [67], the free proline content was determined using the sulfosalicylic acid extraction method [68], and the chlorophyll content was determined using the ethanol-acetone mixture method [69].

### 4.5. RNA Extraction, Sequencing, and Functional Annotation

Total RNA was isolated and purified from tissues sampled at 0 and 6 h with TRIzol reagent (Invitrogen, Carlsbad, CA, USA) according to the manufacturer’s instructions. The quantity and purity of total RNA were determined with a NanoDrop ND-1000 (NanoDrop, Wilmington, DE, USA), and the integrity of total RNA was tested with Agilent2100. A value of the RNA integrity number (RIN) greater than 7.0 was deemed as meeting the qualification requirement. The RNA was then collected selectively through two rounds of mRNA purification with polyadenylate (PolyA) and Oligo (dT) magnetic beads. The collected mRNA was segmented by bivalent cations, and reverse transcriptase was utilized to generate cDNA.

The composite double-stranded DNA and RNA were transformed into double-stranded DNA using DNA polymerase I and RNase H. Simultaneously, dUTP(2′-deoxyuridine 5′-triphosphate) was added to the double-stranded DNA, and the end of the DNA was filled to form a flat end. Then, an A base was added to each end to link with the T base junction, and the fragment was screened and purified using magnetic beads based on its size. The second strand was digested by the UDG enzyme, and a library of fragments with a size of 300 bp (±50 bp) was generated by PCR. Finally, it was sequenced using an Illumina HiSeq 4000 (LC Bio, Hangzhou, China) according to standard procedure, and a length of 150 bp was read.

The original data were further filtered by Cutadapt (https://cutadapt.readthedocs.io/en/stable (accessed on 11 December 2020), version 1.9), and after deleting low-quality and repeated sequences, high-quality clean reads were produced. The clean data were then compared to the alfalfa genome (https://figshare.com/articles/dataset/Medicago sativa genome and annotation files/2623960 (accessed on 11 December 2020)) using HISAT2 (hierarchical indexing for spliced alignment of transcripts)(https://daehwankimlab.github.io/hisat2 (accessed on 11 December 2020), version 2.0.4) to obtain the bam file. The initial assembly of transcripts was accomplished with StringTie (http://ccb.jhu.edu/software/stringtie/ (accessed on 11 December 2020), version 1.3.4; http://ccb.jhu.edu/software/stringtie/ (accessed on 11 December 2020)). Using gffcompare software (http://ccb.jhu.edu/software/stringtie/gffcompare.shtml (accessed on 11 December 2020), version 0.9.8), the test transcript was compared with the reference annotation to generate the final assembly annotation. The ballgown package was utilized for FPKM (Fragments Per Kilobase of exon model per Million mapped fragments) quantification file input. The R package edgeR was used to assess the statistically significant differences between samples. Differentially expressed genes (DEGs) were identified as those with a multiple-fold change of more than 2 (up-regulate) or less than 0.5 (down-regulate) and a *p*-value < 0.05. The biological functions and signaling pathways of DEGs were classified using GO (Gene Ontology) and KEGG (Kyoto Encyclopedia of Genes and Genomes) enrichment analyses.

### 4.6. Validation of Transcriptomic Data

To validate the sequencing quality of the transcriptome data, the selected genes were examined by quantitative real-time reverse transcription PCR (qPCR). Total RNA was reverse transcribed according to the HiScript II 1st Strand cDNA Synthesis kit. Primer 5.0 was used to design the gene-specific primers (Appendix A). The GAPDH gene of alfalfa was used as an internal reference gene. The qPCR was performed as per the directions for ChamQTM Universal SYBR ^®^qPCR Master Mix. Three biological replicates and three identical reactions were performed on each sample. The 2^−△△CT^ comparative approach was utilized to ascertain the expression of particular genes [70]. 

### 4.7. Metabolite Extraction and Metabolic Spectrum Analysis

The metabolites with 50% pre-cooled methanol buffer from tissue samples were taken at 6 h of ST and CK with six biological repetitions. The extraction mixtures were then vortexed for 1 min, incubated for 10 min at room temperature, then kept overnight at −20 °C. The supernatants were collected after 20 min of centrifugation at 4000× *g*. The extractions were then frozen at −80 °C for LC-MS (Liquid Chromatograph Mass Spectrometer, TripleTOF 6600 Plus, SCIEX, UK) analysis. Ten microliters of each extraction mixture were combined to prepare the pooled QC samples. Following machine orders, the LC-MS System gathered all samples. Initially, all chromatographic separations were carried out using ultra-performance liquid chromatography (UPLC) equipment (SCIEX, Macclesfield, UK). An ACQUITY UPLC T3 column (100 mm × 2.1 mm, 1.8 µm, Waters, Cheshire, UK) was utilized for reversed-phase separation. The column oven was kept at 35 degrees Celsius. The injection volume for each sample was 4 μL, with a flow rate of 0.4 mL/min. The column-eluted metabolites were detected using a high-resolution tandem mass spectrometer TripleTOF 6600 Plus (SCIEX, UK). XCMS software (version 3.2) was used to execute the collected MS data pretreatments, including peak picking, peak grouping, retention time correction, second peak grouping, and annotation of isotopes and adducts. The identification of each ion was accomplished by using retention time (RT) and *m*/*z* data. The intensities of each peak were recorded, and a three-dimensional matrix including randomly assigned peak indices (retention time *m*/*z* pairs), sample names (observations), and ion intensity data (variables) was created.

The online KEGG HMDB (the human metabolome database) database was utilized to annotate the metabolites by matching the samples’ exact molecular mass data (*m*/*z*) to those in the database. The metabolite with a mass difference of less than 10 ppm was annotated, recognized, and validated by isotopic distribution studies to determine its chemical formula. The metabolite identification was validated using an in-house fragment spectrum library of metabolites. The intensity of peak data was preprocessed further by metaX (a flexible and comprehensive software for processing metabolomics data). The remaining peaks with missing values were imputed using the k-nearest neighbor approach to improve the data quality further. PCA (principal component analysis) was performed for outlier detection and batch effect evaluation using the preprocessed dataset. The QC data on the injection order were fitted with a robust LOESS (locally weighted scatterplot smoothing) signal correction based on quality control to minimize signal intensity drift over time. In addition, the relative standard deviations of the metabolic characteristics were computed for all QC samples and those with standard deviations greater than 30 percent were eliminated. Student’s *t*-tests were performed to identify variations in metabolite concentrations. Supervised PLS-DA (partial least squares discriminant analysis) was conducted through metaX to discriminate the different variables between groups. The VIP (variable selection) value was calculated. A VIP cutoff value of 1.0 was used to select the most important features.

### 4.8. Integrative Analysis

Following substantial enrichment analysis of GO function and KEGG pathway, all mRNA–metabolomics connection pairs and differentially expressed mRNA–metabolomics relationship pairs were tallied. Afterward, a hypergeometric test was used to identify the functions or pathways significantly enriched in the mRNA–metabolomics connection pairs by comparing them to the background of all mRNA–metabolomics relationship pairings.

### 4.9. Bioinformatics Analysis of Related Genes

ProtParam (http://web.expasy.org/protparam/(accessed on 28 March 2021)) was utilized to predict and analyze the fundamental physical and chemical properties of the protein sequence expressed by the screened genes, including molecular weight, isoelectric point, instability coefficient, fat coefficient, and average hydrophilic index. The program MEME (http://meme-suite.org/tools/meme (accessed on 28 March 2021)) and TBtools were used to evaluate the conserved motifs of proteins. The number of motifs was fixed to 20, and their length was determined to be between 60 and 200 amino acids. 17 DEGs in the phenylpropanoid biosynthesis pathway, of which 7 DEGs were up-regulated and 10 DEGs were down-regulated. 14 DEMs in the phenylpropanoid biosynthesis pathway.

### 4.10. Statistical Analysis

In this study, data were initially processed using Microsoft Excel 2016, and one-way ANOVA was performed to compare the differences between different treatment groups with a significance level of *p*-value < 0.05, using SPSS (version 20.0, SPSS, Inc., Chicago, IL, USA). Figures were drawn using Origin 8.0 (Origin Lab, Northampton, MA, USA) to visualize the results of the data analysis.

## 5. Conclusions

In conclusion, our study demonstrates that foliar spraying of 100 mg/L Na_2_SeO_3_ significantly enhances alfalfa growth, reproductive traits, and physiological characteristics. Through an in-depth integrative analysis of the transcriptome and metabolome, we identified the critical role of phenylpropanoid biosynthesis as a major mechanism behind the observed improvements in alfalfa. Our findings offer valuable insights into the molecular biology of selenium-promoted alfalfa growth and pave the way for further exploration of selenium application in other plant species. This research contributes to the development of effective strategies for improving the productivity and nutritional value of alfalfa and other crops.

## Figures and Tables

**Figure 1 plants-12-02038-f001:**
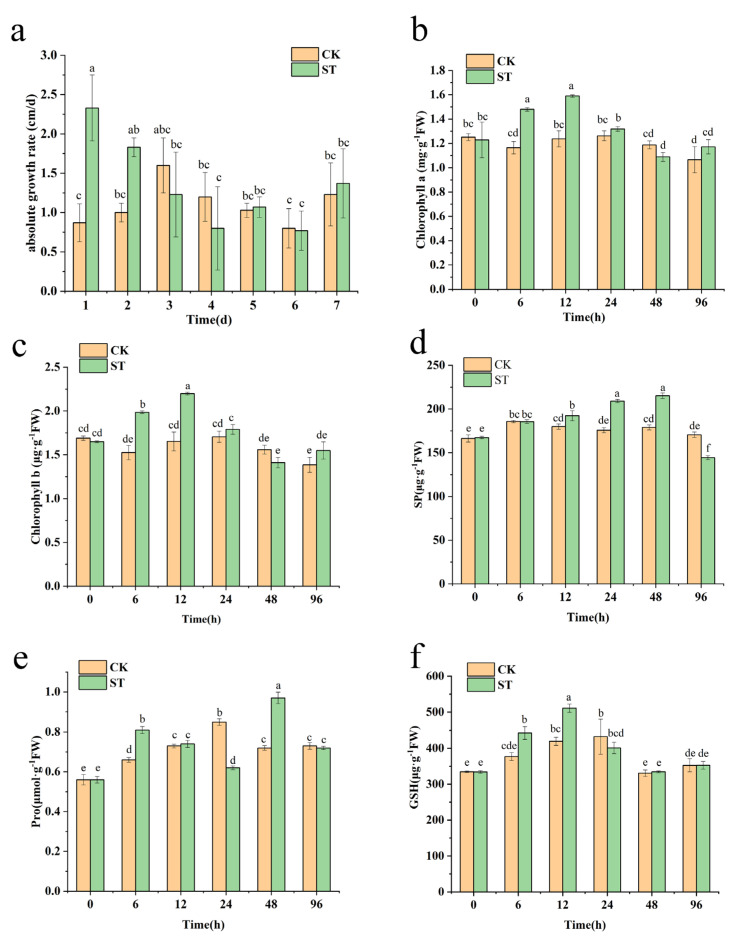
Effects of Na_2_SeO_3_ spraying on the growth rate (**a**) and the content of chlorophyll a (**b**), chlorophyll b (**c**), SP (**d**), Pro (**e**), and GSH (**f**) of alfalfa. SP: soluble protein. Pro: free proline. GSH: reduced glutathione. CK: control group. ST: spraying with 100 mg/L Na_2_SeO_3_. Different letters indicate significant differences (*p*-value < 0.05).

**Figure 2 plants-12-02038-f002:**
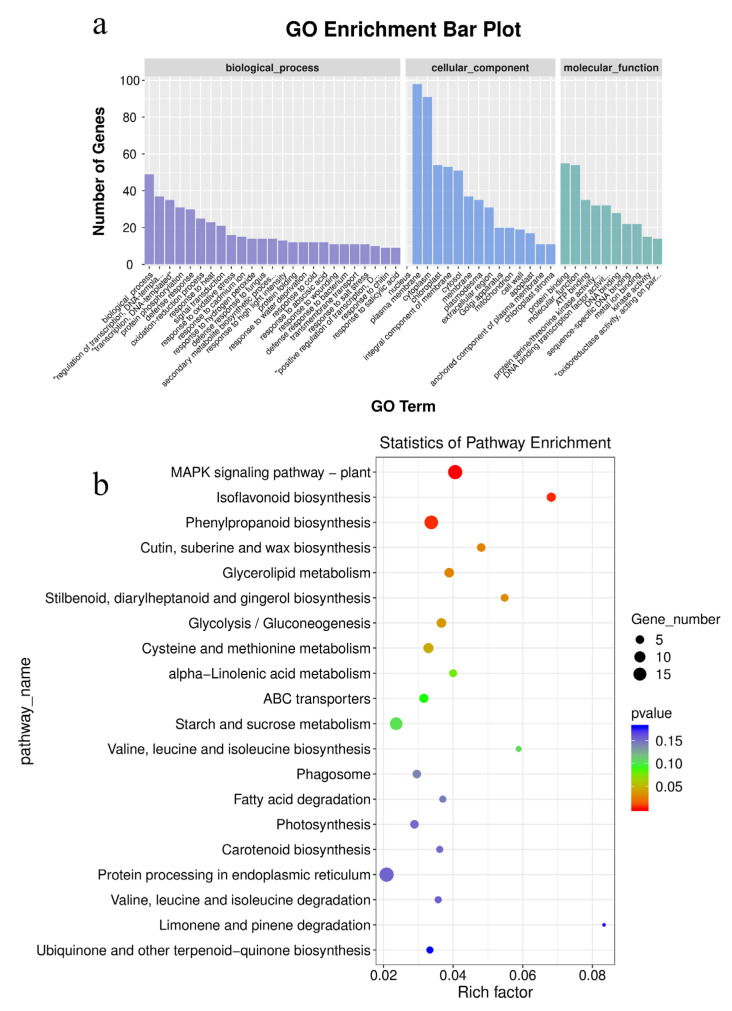
The enrichment analysis of DEGs in GO function (**a**) and KEGG pathway (**b**).

**Figure 3 plants-12-02038-f003:**
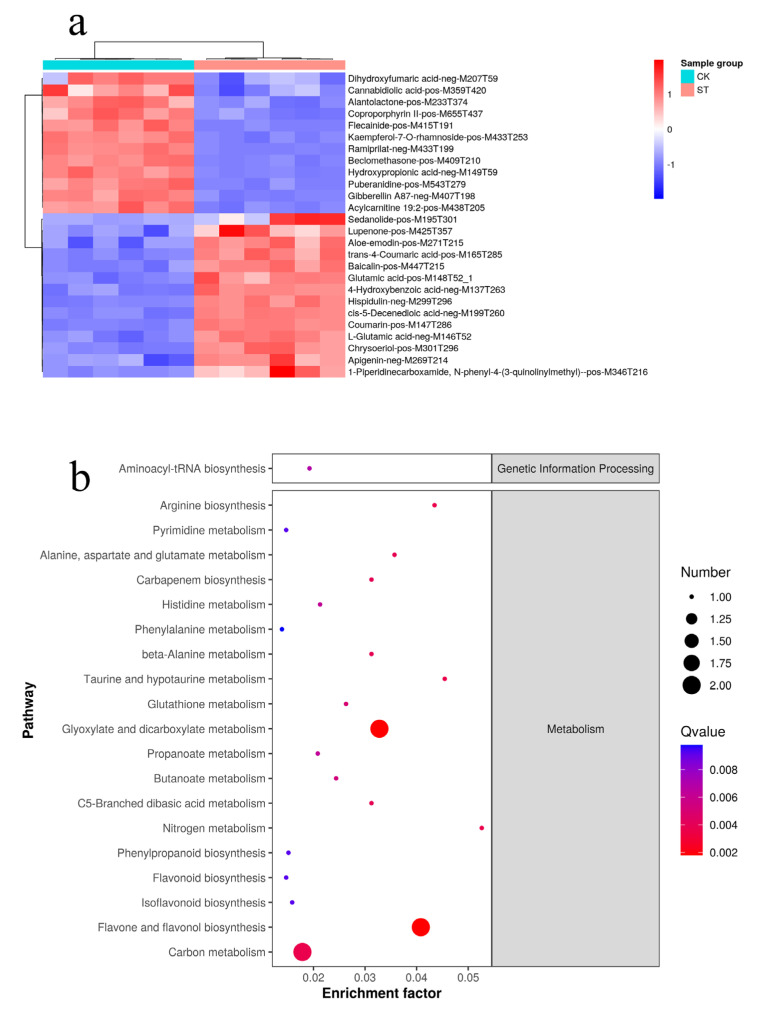
Clustering heatmap (**a**) and KEGG enrichment map (**b**) of DEMs. The size and color of the circles represent the number of genes and *p*-value enriched in the pathway.

**Figure 4 plants-12-02038-f004:**
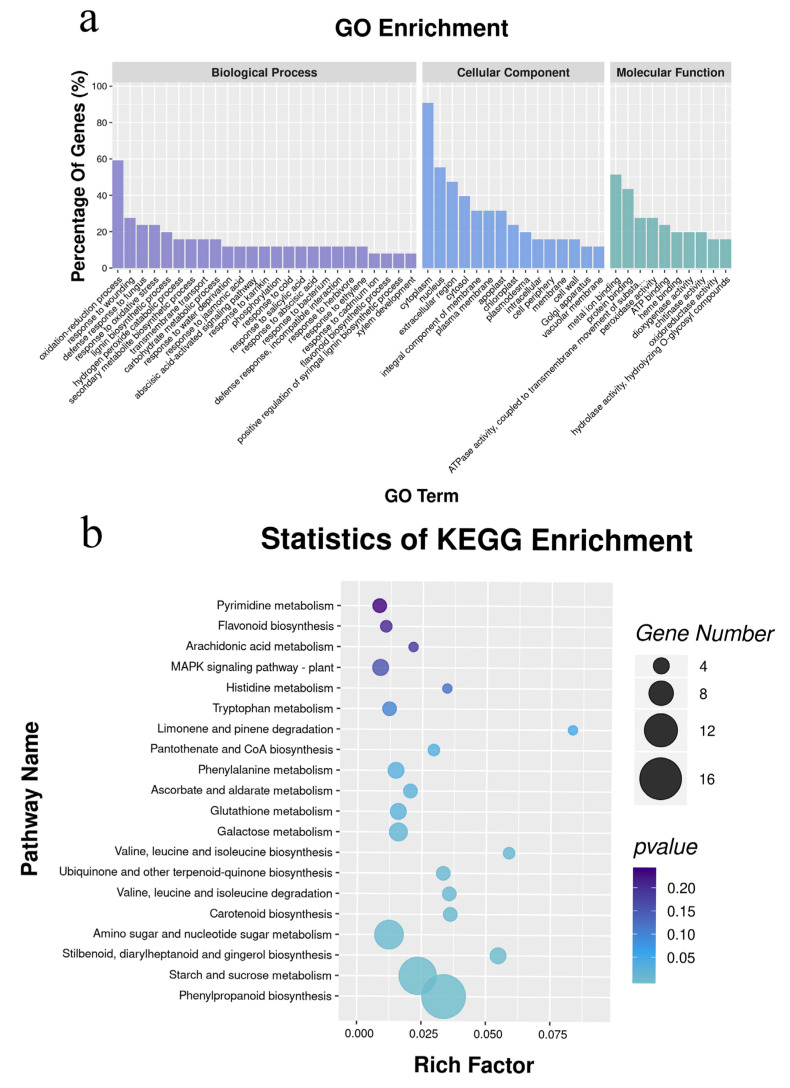
The enrichment analysis in GO function (**a**) and KEGG (**b**) pathway of DEGs filtered via integrative analysis of transcriptome and metabolome.

**Figure 5 plants-12-02038-f005:**
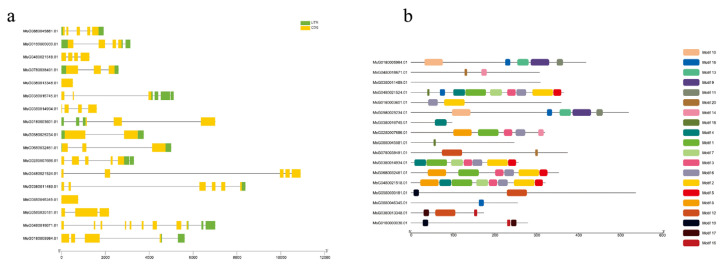
The gene structure (**a**) motif analysis (**b**) of the phenylpropanoid biosynthesis pathway.

**Table 1 plants-12-02038-t001:** Effects of spraying Na_2_SeO_3_ on the plant growth and total Se content of alfalfa at the initial flowering stage.

Treatment	Plant Height/cm	FW/g	DW/g	FW/DW%	Total Se/μg·g^−1^
CK	67.18 ± 2.17	2.06 ± 1.78	0.36 ± 0.02 b	5.88 ± 0.29 a	0.069 ± 0.04 b
ST	68.93 ± 1.98	2.40 ± 1.84	0.58 ± 0.03 a	4.16 ± 0.29 b	0.684 ± 0.11 a

Note: The experiments were carried out in three independent biological replicates, and each replicate contained 10 randomly selected uniform plants. Data were shown as mean ± SD. Different letters in the same column indicated significant differences (*p*-value < 0.05). CK: control group. ST: spraying with 100 mg/L Na_2_SeO_3_. FW: fresh weight. DW: dry weight. FW/DW: fresh weight/dry weight.

**Table 2 plants-12-02038-t002:** Effect of spraying Na_2_SeO_3_ on reproductive traits at the full flowering and podding stages.

Treatment	Number of Florets	Amount of Pollen Per Flower	Pollen Viability%	Pod Spirals/Turns	Seed Numbers Per Pod	1000-Seeds Weight/g
CK	8.60 ± 0.37	23,160 ± 636.87 b	45 ± 0.02 b	2.87 ± 0.09 b	4.93 ± 0.16 b	2.46 ± 0.03
ST	8.83 ± 0.30	33,160 ± 2219.37 a	65 ± 0.05 a	3.45 ± 0.08 a	6.63 ± 029 a	2.45 ± 0.03

Note: The experiments were carried out in three independent biological replicates, and each replicate contained 10 randomly selected uniform plants. Data were shown as mean ± SD. Different letters in the same column indicated significant differences (*p*-value < 0.05). CK: control group. ST: spraying with 100 mg/L Na_2_SeO_3_.

**Table 3 plants-12-02038-t003:** Statistical table of RNA sequencing reads of alfalfa.

Sample	Raw Reads	Raw Bases (G)	Valid Reads	Valid Base (G)	Valid%	Q20 (%)	Q30 (%)	GC (%)
CK1	50,673,258	7.60	49,766,778	7.47	98.21	99.98	97.75	42.50
CK2	49,465,606	7.42	48,475,862	7.27	98.00	99.99	97.75	42.00
CK3	43,909,580	6.59	43,074,702	6.46	98.10	99.99	97.53	41.50
ST1	48,251,932	7.24	47,412,748	7.11	98.26	99.99	97.66	42.50
ST2	46,300,900	6.95	45,163,196	6.77	97.54	99.99	97.62	42.00
ST3	52,649,430	7.90	51,661,074	7.75	98.12	99.99	97.68	42.00

CK: control group. ST: spraying with 100 mg/L Na_2_SeO_3_.

## Data Availability

The RNA-seq datasets generated during the current study have been submitted to the NCBI Sequence Read Archive under the accession number PRJNA908076(https://www.ncbi.nlm.nih.gov/sra/PRJNA908076. Release date: 6 December 2024). The metabolites generated during the current study have been submitted to the MetaboLights under the accession number MTBLS6686 (www.ebi.ac.uk/metabolights/MTBLS6686. Release date: 7 December 2023). Other data supporting the results are included in this published article and its Appendix A.

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
