# Peer review of "Transcriptome and Metabolome Analysis of Selenium Treated Alfalfa Reveals Influence on Phenylpropanoid Biosynthesis to Enhance Growth"

_plants, 2023, doi:10.3390/plants12102038_

Round 1
Reviewer 1 Report
Selenium is an important trace element for human health. This manuscript investigated transcriptome and metabolome changes of alfalfa treated with sodium selenite. The topic is interesting. Selenium-enriched alfalfa would provide selenium for animals and eventually benefits human health. However, I would suggest this manuscript be published in Plants only if being modified substantially.
1. The title is not accurate. Based on the analyses, the authors can speculate that the changes in phenylpropane promote alfalfa growth. However, the current title concluded that the promotion of phenylpropane biosynthesis enhances plant growth, which needs more solid evidence. In addition, in Lines 206-209, the study found that 7 DEGs were up-regulated, 10 DEGs were down-regulated, 7 DEMs were up-regulated, and 7 DEMs were down-regulated in the phenylpropanoid biosynthesis pathway. These findings can not support the conclusion that the promotion of phenylpropane biosynthesis enhances plant growth.
2. Lines 65-71: The reason why foliar spraying was used in this study is unconvincing. The selenium uptake rate of foliar spraying and soil application is more important if selenium waste was taken into consideration.
3. Lines 79-82: The example is not suitable here.
4. Lines 263-265: ‘Zhao found that adding Na2SeO3 to wheat significantly boosted its GSH content. Hence, increased GSH levels in alfalfa seedlings will facilitate Se metabolism.’ Please understand that selenium boosts GSH content does not mean that GSH can facilitate selenium metabolism.
5. There are many minor errors in the manuscript, and a grammar check is a must.
Reviewer 2 Report
In this paper, the authors investigated mechanism of spraying Se on alfalfa growth, and make the conclusion of promotion of phenylpropane biosynthesis to enhance plants growth.
I have some doubts about these. (1) 17 DEGs in the phenylpropanoid biosynthesis pathway, of which 7 DEGs were up-regulated and 10 DEGs were down-regulated. 14 DEMs in the phenylpropane biosynthesis pathway, of which 7 DEMs were up-regulated and 7 DEMs were down-regulated. More upregulated genes than downregulated genes, how to reach the above conclusion.
(2) As shown in figure4, the rich factor of starch and sucrose metabolism is second only to phenylpropanoid biosynthesis. (3) Transcriptome and Metabolome Analysis of the leaves of plants were after treating 6h, and the tested physiology traits were within 96h after treatment. Can these analyses account for reproductive stage traits?
Other minor errors:
1. Line 48, omitted “t” before selenium
2. The last sentence of introduction need to be rewritten.
3. Table 2, the different lower case letters should be added after the SD, just like table 1.
4. Experimental site section: all in the grassland science laboratory? It need not to describe twice.
5. Line 370, 96h after spraying?
6. Sampling at 0, 6, 12, 24, 48, and 96h, and 5 samples per time, only 15 plants were sampling in total?
7. Line180, the description was not consistent with the results in figure3b.
Reviewer 3 Report
Lines 72-84 do not provide relevant information on the subject of the article. It is suggested to focus this topic on examples of transcriptomic analysis in alfalfa or with selenium.
Lines 86-92 it is necessary to rewrite the objective.
Lines 95-96 add the abbreviation ST after the first time it is mentioned, sprayed with Na2SeO3 (ST).
Line 101 remove the period before the parentheses.
Line 134 put the subscripts of Na2SeO3.
Line 263 place the reference number immediately after the author, Zhao [43].
Lines 331-338 in the section "Integrative analysis of transcriptomics and metabolomics" discusses the phenylpropanoid pathway in relation to the interaction of plants with microbes and pathogens. How does the foliar application of Na2SeO3 influence this type of plant-microbe relationship?
All Figures are of poor quality. The clarity is lost when reducing the size of the plots. In Figure 1 it is suggested to change the arrangement of the plots, the form of 3X2 instead of 2x3.
Uniformize the expression P values or p-value.
The proline content and compounds of the phenylpropanoid pathway, such as flavonoids, are also related to an abiotic stress response. Discuss this issue. Could the foliar application of Na2SeO3 be causing stress?
Lines 508-513 the conclusion needs to be rephrased. A comprehensive study was done and it is not reflected in conclusion.
The "Supplementary Materials" results are not integrated into the results and discussion sections.
Round 2
Reviewer 2 Report
All my concerns have been addressed. my suggestion is to accept.